# Biodentine Stimulates Calcium-Dependent Osteogenic Differentiation of Mesenchymal Stromal Cells from Periapical Lesions

**DOI:** 10.3390/ijms26094220

**Published:** 2025-04-29

**Authors:** Mile Eraković, Marina Bekić, Jelena Đokić, Sergej Tomić, Dragana Vučević, Luka Pavlović, Miloš Duka, Milan Marković, Dejan Bokonjić, Miodrag Čolić

**Affiliations:** 1Clinic for Stomatology, Medical Faculty of the Military Medical Academy, University of Defense, 11154 Belgrade, Serbia; erakovic.nino78@gmail.com (M.E.); milosvduka@gmail.com (M.D.); 2Institute for the Application of Nuclear Energy, University of Belgrade, 11080 Belgrade, Serbia; marina.bekic@inep.co.rs (M.B.); sergej.tomic@inep.co.rs (S.T.); luka.pavlovic@inep.co.rs (L.P.); milan.markovic@inep.co.rs (M.M.); 3Institute of Molecular Genetics and Genetic Engineering, University of Belgrade, 11042 Belgrade, Serbia; jelena.djokic@imgge.bg.ac.rs; 4Center for Medical Scientific Information, Faculty of Medicine of the Military Medical Academy, University of Defence, 11040 Belgrade, Serbia; draganavucevic@yahoo.com; 5Medical Faculty Foča, University of East Sarajevo, 73300 Foča, Bosnia and Herzegovina; dejan.bokonjic@ues.rs.ba; 6Serbian Academy of Sciences and Arts, 11000 Belgrade, Serbia

**Keywords:** Biodentine, mesenchymal stromal cells, proliferation, osteogenic differentiation, gene expression

## Abstract

Biodentine, a tricalcium silicate cement, has emerged as a retrograde root-end filling material to promote periapical lesion (PL) healing after apicoectomy. However, its underlying mechanisms remain unclear. This study tested the hypothesis that Biodentine stimulates the osteogenic differentiation of mesenchymal stromal cells (MSCs) derived from PLs. The Biodentine extract (B-Ex) was prepared by incubating polymerized Biodentine in RPMI medium (0.2 g/mL) for three days at 37 °C. B-Ex, containing both released microparticles and soluble components, was incubated with PL-MSCs cultured in either a basal MSC medium or suboptimal osteogenic medium. Osteogenic differentiation was assessed by Alizarin Red staining and the expression of 20 osteoblastogenesis-related genes. Non-cytotoxic concentrations of B-Ex stimulated the proliferation of PL-MSCs and induced their osteogenic differentiation in a dose-dependent manner, with a significantly enhanced effect in suboptimal osteogenic medium. B-Ex upregulated most early and late osteoblastic genes. However, the differentiation process was prolonged, as indicated by the delayed expression of wingless-type MMTV integration site family member 2 (WNT2), bone gamma-carboxyglutamate protein (BGLAP), bone morphogenic protein-2 (BMP-2), growth hormone receptor (GHR), and FOS-like 2, AP-1 transcription factor subunit (FOSL2), compared with their expression under optimal osteogenic conditions. The stimulatory effect of B-Ex was primarily calcium dependent, as it was reduced by 85% when B-Ex was treated with the calcium-chelating agent EGTA. In conclusion, Biodentine promotes the osteogenic differentiation of PL-MSCs in a calcium-dependent manner, supporting its stimulatory role in periapical healing.

## 1. Introduction

Biodentine, a trade name for an innovative bioactive dentine substitute, was developed in 2009 by Septodont, Saint-Maur-des-Fossés, France. It belongs to the group of mineral trioxide aggregate (MTA) cements, which have been extensively utilized in various pulp and endodontic procedures. Biodentine is composed of tricalcium silicate (80.7%), calcium carbonate (14.25%), and zirconium dioxide (5%) to enhance radiopacity. Compared with other MTA cements, it offers a faster setting time (6–12 min) and superior mechanical properties. Its hardening process involves hydration, a silicate gel, and calcium hydroxide formation, with porosity gradually decreasing over time [1,2].

Biodentine forms a crystalline network with hard dental tissues, releasing calcium and hydroxide ions, which stimulate dentin formation and osteoblast activity. It is widely used in dentistry for dentin replacement, pulp capping, perforation repair, pulpotomy, treatment of resorptions, or apexification. Its key properties include hermetic sealing, reduction in microleakage, and prevention of bacterial infiltration. Contact with dentin promotes hydroxyapatite deposition in dentinal tubules, enhancing chemical bonding and protection against demineralization. Biodentine is stable in moist environments and resistant to disintegration due to polymer additives. Its high pH (~12) provides antimicrobial properties and acts as a disinfectant [1,2].

Numerous studies have shown that Biodentine is a biocompatible material [3,4], although a certain degree of cytotoxicity has been observed when cells are cultured on Biodentine discs for extended periods or when Biodentine powder is used in high concentrations [5,6].

Biodentine’s biological activity has been primarily evaluated through experimental models designed for its original indications, such as pulp repair, classifying it as a bioactive endodontic cement. Studies comparing Biodentine and MTA using primary human pulp cells showed that both materials promote cell proliferation, differentiation, migration, and mineralization with no significant differences in biological potential [7,8]. Additionally, studies on osteoblast cell lines and periodontal ligament stem cells showed that Biodentine promotes osteoblast differentiation, increases the expression of genes associated with osteoblast activation or differentiation, and induces hydroxyapatite crystal cluster formation [9,10,11,12].

Due to its ability to stimulate osteoblasts and its effective sealing properties in the root canal, Biodentine has also been used in apical surgery. However, its effectiveness in this area is not sufficiently known. Current evidence is limited to positive findings from one clinical study [13] and several clinical cases demonstrating its effect on the healing of periapical lesions (PLs) after apicoectomy and retrograde root-end filling [14,15]. Although larger clinical studies are required to evaluate the effectiveness of Biodentine in root-end filling and its potential disadvantages, such as difficulty in initial handling [14], the experience of the apical surgeon is of crucial importance.

Our previous studies showed that Biodentine possesses immunomodulatory properties by suppressing pro-inflammatory cytokines and osteolytic mediators while augmenting anti-inflammatory cytokines in cultures of cells infiltrating PLs [16]. Based on these findings and the positive results of its application in apical surgery, we hypothesized that Biodentine additionally stimulates regenerative processes in periapical tissue through the induction of osteoblast differentiation of mesenchymal stromal cells (MSCs). The hypothesis was evaluated in MSC cultures derived from PLs (PL-MSCs), which were cultivated with different concentrations of an extract obtained by incubating Biodentine in a culture medium. In addition, the role of calcium (Ca) in these processes was investigated. We believe this unique study model is suitable for drawing valid conclusions, as leachable components and microparticles of Biodentine during root-end filling could reach periapical tissues and affect MSC differentiation. Therefore, this in vitro model best mimics the process of osteogenic differentiation in vivo during periapical healing.

## 2. Results

### 2.1. Establishment and Characterization of PL-MSCs

Four MSC lines were successfully established from clinically asymptomatic PLs. Each line was analyzed for its clonogenic potential, growth characteristics in culture, and morphological changes during passaging. Following four passages, the phenotypic profile was assessed, and the cells’ ability to differentiate into osteoblasts, chondroblasts, and adipocytes was evaluated. One cell line was used to assess the effect of B-Ex on cytotoxicity and the proliferation of PL-MSCs, three cell lines were used to assess their phenotypic properties, and two cell lines were employed to investigate the effect of B-Ex on the osteogenic differentiation of PL-MSCs. One cell line was used to study the role of Ca in osteogenic differentiation.

PL-MSCs exhibited a fibroblast-like morphology characteristic of MSCs derived from other tissues (Figure 1A). They also demonstrated clonogenic potential, as evidenced by their ability to form colony-forming-unit fibroblast (CFU-F) colonies (Figure 1B). All PL-MSCs displayed the capacity to differentiate into osteoblasts, chondroblasts, and adipocytes when cultured in the appropriate induction media. However, they exhibited a higher differentiation potential toward osteoblasts and chondroblasts compared with adipocytes. The histochemical analysis of the respective cultures is presented in Figure 1C.

The phenotypic characteristics of PL-MSCs pooled from three cell lines are summarized in Table 1, while fluorescence histograms of a representative cell line are shown in Appendix A. The results show that all PL-MSCs expressed CD73, CD90, and CD166, while approximately 90% of the cells were positive for CD105 and CD39. A significantly lower percentage of cells expressed CD146 (28.18 ± 6.25%) and CD56 (20.04 ± 4.88%). The lowest expression levels were observed for STRO-1 (8.06 ± 4.60%) and SSEA4 (12.12 ± 5.56%). As anticipated, PL-MSCs did not express markers associated with hematopoietic stem cells (CD34), total hematopoietic cells (CD45), monocytes (CD14), T lymphocytes (CD3), or B lymphocytes (CD19), as less than 2% of the cells were positive for these markers.

### 2.2. Cytotoxicity of the Biodentine Extract

B-Ex was prepared as described in Section 4. The extract contained microparticles of Biodentine and had a pH value of 11.5 and a total Ca content of 828.33 mg/L (20.68 mM). Before use in subsequent experiments, the pH of B-Ex was adjusted to 7.4. The PL-MSCs were treated with different concentrations of B-Ex for 24 and 72 h, after which the metabolic activity of the cultures was assessed using the MTT assay.

The results presented in Figure 2A show that extract concentrations of 50% and higher significantly decreased cell viability at 72 h in a concentration-dependent manner (*p* < 0.01 and *p* < 0.001, respectively). After 24 h, the concentrations of 75% and 100% B-Ex were cytotoxic. In contrast, lower concentrations (10%, and 30%) increased the metabolic activity of PL-MSCs, particularly after 72 h, in a concentration-dependent manner (*p* < 0.05 and *p* < 0.001, respectively). The inhibition of MTT activity resulted from cell death, as confirmed by propidium iodide (PI) staining (Figure 2B), whereas the increased MTT activity was due to enhanced cell proliferation, as demonstrated by the increased incorporation of [^3^H]-thymidine in the presence of 30% B-Ex (*p* < 0.05) (Figure 2C).

### 2.3. The Effect of Biodentine Extract on Osteogenic Differentiation of PL-MSCs

Based on the results of the proliferation assay, PL-MSCs were cultured with 10% and 30% B-Ex in the basal medium without any osteogenic stimuli. In addition, these concentrations of the extract were added to the PL-MSCs, which were cultured either with the complete osteogenic medium or with basal medium supplemented with 30% osteogenic medium, referred to as the suboptimal osteogenic medium. After 21 days, the cultures were stained with Alizarin Red to visualize mineralized islands characteristic of osteoblasts.

The results presented in Figure 3A show that both concentrations of B-Ex significantly enhanced osteoblastogenesis in the suboptimal osteogenic medium (*p* < 0.05 and *p* < 0.001, respectively), as did the higher concentration of B-Ex in the control basal medium (*p* < 0.001). No significant modulation of osteoblastic differentiation was observed in the optimal osteogenic medium, in which osteogenic differentiation was highest. Figure 3B shows representative images of cultures stained with Alizarin Red.

### 2.4. The Biotine Extract Significantly Modulates the Expression of Osteoblastic Genes

The next phase of the study aimed to evaluate the impact of two concentrations of B-Ex (10% and 30%) on the expression of 20 genes during the differentiation of PL-MSCs into osteoblasts, which were cultured in either basal medium or suboptimal osteogenic medium. Gene expression in PL-MSC cultures grown in the complete osteogenic medium served as a positive control. Since previous experiments indicated that B-Ex had no modulating effect in the complete osteogenic medium, these cultures were not treated with B-Ex. Gene expression was analyzed at day 5 (early phase) and day 16 (late phase) of osteogenic differentiation. All genes were analyzed at both phases of osteoblast differentiation.

### 2.5. Genes Expressed in Early and Late Phases of Osteogenic Differentiation

The expression of four early osteogenic differentiation genes, runt-related transcription factor 2 (*RUNX2*), Osterix (*SP7*), wingless-type MMTV integration site family member 2 (*WNT2*), and collagen type I alpha 1 chain (*COL1A1*), was evaluated, with the results shown in Figure 4. All genes were significantly upregulated (*p* < 0.001) in the early phase when PL-MSCs were cultured in the complete osteogenic medium (positive control), followed by a notable decline in expression in the late phase. A similar trend was observed in PL-MSCs treated with 30% B-Ex in a suboptimal osteogenic medium (*p* < 0.001). Higher concentrations of B-Ex in the basal medium upregulated *RUNX2*, *SP7*, and *COL1A1*, with a more pronounced effect in the late phase (*p* < 0.001). In contrast, *WNT2* was downregulated during the early phase (*p* < 0.01). Lower B-Ex concentrations had a pronounced upregulating effect on the *WNT2* expression in the suboptimal osteogenic medium, especially in the late phase (*p* < 0.001), and the opposite suppressive effect in the basal medium (*p* < 0.01).

The expression of three genes predominantly active during the late phase of osteogenic differentiation, bone gamma-carboxyglutamate protein (*BGLAP*), alkaline phosphatase (*ALP*), and bone morphogenetic protein-2 (*BMP-2*), was evaluated. As shown in Figure 4, *BGLAP* and *ALP* exhibited significantly higher expression in the late phase in the complete osteogenic medium (*p* < 0.001) compared with the early phase, whereas *BMP-2* showed the opposite trend. However, in the presence of 30% B-Ex within a suboptimal osteogenic medium, all three genes were upregulated during the late phase (*p* < 0.001). Among them, *BMP-2* was also significantly upregulated at lower B-Ex concentrations (*p* < 0.001). A similar expression pattern was observed for both *BGLAP* (*p* < 0.001) and *ALP* (*p* < 0.01) in the basal medium with both B-Ex concentrations.

### 2.6. Cytokine/Cytokine Receptor Genes and Hormone Receptor Genes

The expression of three cytokine genes, transforming growth factor beta (*TGF-β*), fibroblast growth factor 2 (*FGF2*), and connective tissue growth factor (*CTGF*), was evaluated (Figure 5). *TGF-β* and *FGF2* were significantly upregulated in the complete osteogenic medium during both phases, with a stronger effect in the late phase (*p* < 0.001). A similar pattern was observed in the 30% B-Ex/suboptimal osteogenic medium (*p* < 0.001 and *p* < 0.05, respectively). Conversely, *CTGF* showed an opposite pattern of expression (upregulation only in the early phase; *p* < 0.001), with lower concentrations of B-Ex further decreasing its expression in the late phase (*p* < 0.05). However, in the basal medium, neither concentration of B-Ex significantly altered the expressions of the three genes.

The expression of interleukin 6 signal transducer (*IL-6ST*) mRNA was significantly upregulated only in the early phase in the complete osteogenic medium (*p* < 0.001) and 30% B-Ex/suboptimal osteogenic medium (*p* < 0.05). In the osteogenic medium, the growth hormone receptor gene (*GHR*) was expressed only in the early phase of osteogenic differentiation (*p* < 0.001). Higher concentrations of B-Ex added to both the suboptimal osteogenic and basal medium exerted the same stimulatory effect, but only in the late phase (*p* < 0.001), as did the lower concentration of B-Ex in the suboptimal osteogenic medium (*p* < 0.05). Upregulation of the parathyroid hormone receptor gene (*PTHR*) was seen in the late phase of osteogenic differentiation in the osteogenic and 30% B-Ex/basal medium (*p* < 0.001). In contrast, an increase in *PTHR* expression in 30% B-Ex/suboptimal osteogenic medium was noticed in the early phase (*p* < 0.05) (Figure 5).

### 2.7. Genes Modeling Bone and Extracellular Matrix

Three genes modeling the bone and extracellular matrix, receptor activator of nuclear factor kappa-B ligand (*RANKL*), cathepsin K (*CTSK*), and versican (*VCAN*), were analyzed (Figure 6). In the complete osteogenic medium, all genes were upregulated only in the early phase of osteoblastic differentiation, followed by a decrease to basal levels (*RANKL* and *CTSK*) or complete suppression (*VCAN*) (*p* < 0.01) in the late phase. Higher concentrations of B-Ex increased the expression of *RANKL* in the early phase in both the control basal medium (*p* < 0.05) and the suboptimal osteogenic medium (*p* < 0.01). In contrast, neither concentration of B-Ex modulated the expression of *CTSK* and *VCAN*, regardless of the medium used.

### 2.8. Genes Modulating Osteoblast Signaling

Four genes involved in osteoblast signaling, FosB proto-oncogene, AP-1 transcription factor subunit (FOSB), FOS-like 2, AP-1 transcription factor subunit (FOSL2), discoidin domain receptor tyrosine kinase 1 (DDR1), and discoidin domain receptor tyrosine kinase 2 (DDR2), were evaluated. The results are presented in Figure 7. The expression of FOSB was increased in the complete osteogenic medium and 30% B-Ex/suboptimal osteogenic medium during the late phase of osteoblast differentiation (*p* < 0.001). However, the same concentration of B-Ex had an inhibitory effect in the basal medium. The expression of FOSL2 was downregulated during both investigated phases of osteoblastic differentiation (*p* < 0.001). However, 30% B-Ex had a stimulatory effect in both the suboptimal osteogenic and basal medium (*p* < 0.001) in the late phase. In contrast, FOSL2 expression in the PL-MSCs cultured in the basal medium was significantly downregulated with both concentrations of the extracts at the early phase of osteogenic differentiation (*p* < 0.001).

The expression of DDR1 was significantly upregulated in the complete osteogenic medium (*p* < 0.001) during the early phase of osteoblastic differentiation, followed by a significant decline thereafter. A similar stimulatory effect was observed during the early phase with both concentrations of B-Ex in the basal medium (*p* < 0.001), as well as with 30% B-Ex in the suboptimal osteogenic medium (*p* < 0.001). The expression of DDR2 was lower than that of DDR1, but followed a similar pattern—upregulation in the early phase of PL-MSCs differentiation in the presence of 30% B-Ex (both types of medium) as well as in the complete osteogenic medium (*p* < 0.001).

### 2.9. The Role of Calcium in PL-MSC Proliferation and Osteoblastic Differentiation Induced by Biodentine Extract

To investigate the role of Ca ions in the PL-MSC proliferation and osteoblastic differentiation induced by B-Ex, we used EGTA, a Ca-chelating agent. PL-MSCs cultivated in the complete basal medium were treated with either 30% B-Ex alone or 30% B-Ex with 2 mM EGTA for 24 h. After this incubation, the extract and extract/EGTA were washed away, and the cells were further cultured for an additional 48 h. Cell proliferation and RUNX2 expression were then assessed. A parallel set of cultures was treated with 6.2 mM Ca (matching the concentration determined in 30% B-Ex), with and without EGTA, as a specific control.

The results, presented in Table 2, show that a 24 h exposure to CaCl_2_ was sufficient to trigger both proliferation and *RUNX2* expression in PL-MSCs (*p* < 0.001), and these effects were completely inhibited by EGTA. Proliferation and RUNX2 expression were also increased in PL-MSC cultures treated with 30% B-Ex (*p* < 0.001), and both parameters were slightly higher compared with Ca-supplemented cultures. However, a statistically significant difference (*p* < 0.05) was observed only at the level of *RUNX2* expression. EGTA completely suppressed B-Ex-induced cell proliferation, while the suppression of *RUNX2* expression was approximately 85%.

## 3. Discussion

Biodentine has recently been investigated as a root-end filling material in endodontic surgery [13,14,15]. However, it remains unclear whether its beneficial effects on periapical wound healing are primarily due to improved root obturation or its additional support for osteogenesis. To test this hypothesis, we prepared an extract of this cement (B-Ex) and used it to study its effect on the differentiation of PL-MSCs into osteoblasts. This original in vitro model mimics the impact of Biodentine’s released compounds on the osteoinductive potential of this population of inflammatory MSCs [13,14,15], which come into close contact in vivo with Biodentine and its components.

We showed that PL-MSCs satisfy all the characteristics of MSCs as defined by Dominici et al. [17], including fibroblastoid morphology; adherence to plastic substrates; clonogenicity; differentiation potential into osteoblasts, chondroblasts, and adipocytes; as well as the expression of characteristic MSC markers. The PL-MSCs highly or moderately expressed CD73, CD90, CD105, CD166, CD39, CD146, and CD56 but exhibited low levels of STRO-1+ and SSEA4+ cells, similar to the PL-MSCs we previously established [18]. Some specific characteristics are most probably related to the fact that PL-MSCs established from an inflamed microenvironment originate from both residual periodontal ligament cells (PDLSCs) and pericytes from newly formed vasculature [18] within the granulomatous tissue of PLs. Compared with PL-MSCs, PDLSCs exhibit lower expressions of CD105 and CD166 [19], higher expressions of STRO-1 [20] and other stem cell markers [21], as well as an increased differentiation potential toward adipocytes [20,22].

The initial screening tests aimed to determine non-cytotoxic concentrations of B-Ex for further studies. Although Biodentine has been reported to be non-cytotoxic, as documented in studies on L929 cells [23,24], dental pulp fibroblasts [25], odontoblasts [26], osteoblasts [27], and SHED MSCs [4], the results of our study, performed strictly according to ISO 10993-5 [28], showed that concentrations of 50% B-Ex and higher are cytotoxic. These findings align with those of previous studies investigating the effects of a powder prepared from polymerized and hardened Biodentine [5] or a conditioned medium of Biodentine [6] prepared under similar conditions as in our study. The cytotoxic effect may be attributed to increased concentrations of Ca or other ions and/or microparticles released from Biodentine during the extraction procedure [4,5,28]. Detailed SEM, EDX, FTIR, and XRD analyses of both the supernatants and microparticles from B-Ex are currently in progress. An increased pH value, as suggested by other authors [5], is not relevant, since in our study, the pH of the extract was adjusted to physiological values.

The key finding of this study is that B-Ex stimulates PL-MSC proliferation and osteoblast differentiation in complete α-MEM, further enhancing osteoblastogenesis in the 30% osteoinductive medium. However, these effects were less pronounced compared with the complete osteogenic medium. MSC differentiation into osteoblasts is a complex process involving extracellular and hormonal interactions that activate intracellular signaling pathways and transcription factors [29,30]. RUNX2 plays a central role by regulating key pathways (FGF, Hedgehog, WNT, Pthlh) and transcription factors (SP7, Dlx5), promoting osteoblast differentiation while inhibiting chondrogenesis [31,32]. RUNX2 is crucial for mesenchymal stem cell commitment to the osteoblast lineage. It drives early osteoblast differentiation but inhibits full maturation, balancing osteoprogenitor proliferation and differentiation. By regulating bone matrix protein expression and interacting with key factors, RUNX2 ensures proper skeletal development [31,32]. Canonical WNT signaling, particularly WNT2, activates *RUNX2* and *SP7* transcription via β-catenin [32,33], providing the rationale for our gene expression analysis.

SP7, or Osterix, is a zinc finger transcription factor encoded by the *SP7* gene that is essential for osteoblast differentiation and bone formation. Acting downstream of RUNX2, it drives committed precursors toward full osteoblast maturation. The critical role of SP7 is evident in SP7-deficient mice, which completely lack bone formation. SP7 directly regulates key osteoblast genes, including those for type I collagen, osteocalcin, and bone sialoprotein, ensuring proper bone matrix formation and mineralization. The RUNX2–SP7 axis is central to the activation of osteogenic genes [34].

WNT2, a member of the WNT family, plays a key role in bone formation through the Wnt/β-catenin signaling pathway. This pathway promotes mesenchymal stem cell differentiation into osteoblasts, enhancing osteogenic gene expression and increasing bone mass. Inhibition of Wnt signaling reduces osteoblast activity, leading to bone loss. Dysregulation of this pathway is linked to low bone mass and a higher risk of fractures [35].

In the osteoinductive medium, *RUNX2* peaked early (40-fold increase), with moderate *SP7* and *WNT2* expression. In the late phase, *RUNX2* and *SP7* declined, while *WNT2* was suppressed. A similar but weaker pattern appeared under suboptimal osteogenic conditions in the presence of higher concentrations of B-Ex, confirming the primary role of *WNT2* in regulating the other two genes. However, this hypothesis was not supported when B-Ex was used without additional osteogenic stimuli, as the increased expression of *RUNX2* and *SP7* was followed by suppressed *WNT2* expression. Moreover, lower concentrations of B-Ex in a suboptimal osteogenic medium prolonged *WNT2* expression until the late phase without affecting *RUNX2* and *SP7*. Based on these findings, it can be postulated that suboptimal osteogenic stimulation prolongs osteogenic differentiation of PL-MSCs and modifies osteogenic signaling in these cells. However, this presumption requires further verification.

Three key genes involved in matrix mineralization are *BGLAP*, *COL1A1*, and *ALP*. *BGLAP* encodes osteocalcin (OCN), a late osteoblast marker essential for bone mineralization and remodeling [36]. OCN binds calcium and hydroxyapatite, facilitating mineral deposition, while its soluble form acts as a hormone in various physiological processes [36,37]. It enhances bone formation, osteoblast adhesion, and bone healing around hydroxyapatite-based composites relevant to Biodentine [38,39]. We observed consistently higher *BGLAP* expression in the late differentiation phase across all conditions, with B-Ex further enhancing its levels beyond those in the complete osteogenic medium.

*BGLAP* expression is regulated by factors such as parathyroid hormone (PTH) [37,40]. We detected increased *PTHR* expression in the late differentiation phase, coinciding with *BGLAP* upregulation. *PTHR*, a G-protein-coupled receptor for PTH and PTH-related peptides, plays a crucial role in osteoblast differentiation and proper bone formation. The signaling involves the cAMP–PKA and calcium–PKC pathways. B-Ex-induced *PTHR* upregulation may contribute to calcium homeostasis and bone remodeling by promoting osteoblast proliferation, reducing apoptosis, counteracting PPARγ inhibition, and enhancing bone formation [41]. A similar pattern was observed for *GHR* gene expression. GHR is a membrane receptor for growth hormone that mediates its anabolic effects on bone by stimulating osteoblast proliferation, differentiation, and matrix deposition. It plays a crucial role in longitudinal bone growth and remodeling, primarily through insulin-like growth factor-1 (IGF-1), which enhances osteoblast activity and matrix formation [42]. The B-Ex stimulatory effect, particularly in the basal medium, suggests that this bioactive cement promotes mineralization by regulating *GHR* and *PTHR* expression.

*COL1A1* expression peaked early in the complete osteoinductive medium before declining, following a similar pattern under suboptimal conditions. B-Ex enhanced its early expression in suboptimal conditions and increased its late-phase expression in the control medium, paralleling *RUNX2*. Type I collagen is the most abundant collagen in the human body and a key component of the bone matrix, constituting 90% of bone proteins [43]. It provides structural support and serves as a framework for mineral deposition. The *COL1A1* gene encodes the alpha-1 chain of type I collagen, and its expression is a key marker of osteoblast activity. Type I collagen is regulated by *RUNX2* and *SP7* via p38 and ERK signaling [43]. While both *BGLAP* and *COL1A1* typically rise as osteoblasts mature, we confirmed this only for *BGLAP*, possibly due to timing differences or PL-MSC-specific traits.

Unlike *COL1A1*, *ALP* followed *BGLAP*’s late-phase increase across all conditions, with B-Ex further stimulating its expression. ALP is an enzyme expressed by osteoblasts that plays a critical role in bone mineralization. It hydrolyzes phosphate esters, increasing the local phosphate concentrations necessary for hydroxyapatite crystal formation, which is essential for bone hardness and strength. ALP is crucial for calcification and a key marker of dental MSC differentiation and bone regeneration [44]. Though often considered an early osteoblast marker, its higher late-phase expression in our study suggests prolonged differentiation and mineralization, aligning with other gene expression patterns in PL-MSC cultures with B-Ex but also in cultures with the complete osteogenic medium.

BMP-2, a late marker of osteoblast differentiation, belongs to the BMP cytokine family within the TGF-β superfamily. Originally isolated from the mineralized bone matrix, BMP-2, along with BMP-4, BMP-5, and BMP-7, promotes MSC osteoblast differentiation and bone formation, with BMP-2 exhibiting the highest activity [45]. It binds to membrane receptors, activating the Smad pathway to regulate *RUNX2, SP7*, and *ALP* expression [46]. Our results indicate increased *BMP-2* expression in the early phase of PL-MSC differentiation in a complete osteogenic medium followed by inhibition in the late phase. However, in B-Ex cultures, BMP-2 expression was stimulated in the late phase, suggesting again that B-Ex prolongs osteoblastic differentiation under suboptimal conditions.

TGF-β mRNA expression exhibited an inverse pattern, with higher levels in the early phase under suboptimal osteogenic conditions and in the late phase under optimal conditions. B-Ex stimulated both cytokines in the late differentiation phase across control and suboptimal cultures. This suggests that BMP-2 is more crucial in early osteogenesis, while TGF-β dominates later, at least in our model. TGF-β, a multipotent cytokine, regulates proliferation, differentiation, migration, adhesion, and extracellular matrix synthesis [47]. In bone, it is produced by osteoblasts and osteocytes, promoting MSC migration and early osteoblast differentiation while inhibiting late differentiation and enhancing proliferation and mineralization. TGF-β signals through Smad-dependent and non-Smad pathways. In the canonical pathway, TGF-β binds to its receptors, triggering R-Smad phosphorylation. R-Smads then complex with co-Smad and enter the nucleus to regulate gene expression. Non-Smad pathways, including MAPK signaling, also influence osteoblast function and bone formation. Conditional knockout models lacking TGF-β receptors in osteoblasts exhibit impaired bone formation, highlighting the importance of TGF-β in maintaining bone mass [47]. The effects of TGF-β on MSC osteoblastic differentiation depend on its concentration, culture conditions, MSC type, and cell density [48].

B-Ex also upregulated FGF2 expression in both culture conditions, with a stronger effect in the late osteoblastic differentiation phase. As a key member of the FGF family, FGF-2 regulates bone formation and osteoblast differentiation [49] while stimulating osteoprogenitor proliferation through the MAPK, PI3K/Akt, and ERK pathways. Beyond bone formation, FGF-2 participates in angiogenesis, wound healing, morphogenesis, and metabolism [50]. It promotes the expression of *RUNX2*, *COL1A1*, *ALP*, and *BGLAP*, facilitating extracellular matrix deposition [51]. The observed positive correlation between *FGF2* and *BGLAP* as well as *AP* suggests that Biodentine’s effect on *FGF2* expression is more relevant for bone mineralization than for other functions.

CTGF is a matricellular protein from the CCN family that regulates osteoblast differentiation by promoting extracellular matrix (ECM) production, osteoblast proliferation, and bone matrix protein expression such as collagen and osteocalcin. It is involved in key signaling pathways, including TGF-β and Wnt, essential for bone formation and remodeling [52]. Our findings show increased *CTGF* expression in the early phase of osteogenic differentiation, both in standard and suboptimal osteogenic cultures supplemented with 30% B-Ex, suggesting its primary role in ECM stimulation. This aligns with the upregulation of *VCAN*, *CTSK*, and *RANKL*, though this correlation was significant only under optimal conditions.

*VCAN* encodes a chondroitin sulfate proteoglycan that supports early osteoblast differentiation by facilitating MSC adhesion. It regulates ECM assembly and organization, which is critical for osteoblast attachment. VCAN also influences signaling through Wnt/β-catenin and TGF-β pathways, which are vital for osteoblastogenesis [53]. CTSK, a lysosomal cysteine protease, primarily functions in osteoclasts, but it also contributes to osteoblast matrix turnover and osteoblast–osteoclast communication. It plays a crucial role in bone resorption by degrading type I collagen and other components of the bone extracellular matrix. CTSK is essential for normal bone remodeling, and its dysregulation is linked to bone disorders such as osteoporosis and pycnodysostosis [54]. Similarly, RANKL activates osteoclasts via RANK binding but also plays a role in MSC differentiation, particularly in early osteogenesis. It can enhance osteogenesis by stimulating signaling pathways such as NF-κB and MAPK, which are involved in bone formation. RANKL–RANK signaling can indirectly support osteoblast differentiation by promoting osteoclast activity, which releases bone-derived growth factors that stimulate osteoblast function [55]. In cultures with Biodentine, our results did not support a major role for VCAN and CTSK under suboptimal osteogenic conditions except for RANKL, suggesting that other genes may be more relevant or that the timing of gene expression analysis was suboptimal.

IL-6ST is a coreceptor in the signaling pathways of the IL-6 cytokine family, including IL-6, IL-11, and others. The gene is vital for maintaining bone homeostasis and participates in both anabolic and catabolic bone responses. By activating IL-6ST, IL-6 facilitates the early commitment of MSCs to the osteoblast lineage, as well as osteoblast proliferation and survival, through the JAK/STAT3, MAPK, and PI3K/Akt pathways. Additionally, IL-6 upregulates key osteogenic genes, such as *RUNX2*, *WNTs*, and *BMPs*. In this context, IL-6ST can be classified as an early osteoblastic gene. However, it also functions as a late osteoblastic gene, as its activity persists in mature osteoblasts and osteocytes, supporting bone remodeling and matrix turnover through osteoblast–osteoclast interactions [56]. Our findings suggest that while *IL-6ST* is primarily expressed in the early phase of osteogenic differentiation, its activation requires stronger osteogenic stimuli, as Biodentine alone is insufficient.

DDR1 and DDR2, collagen-binding receptor tyrosine kinases (RTKs), play essential roles in osteoblast adhesion, migration, and differentiation. DDR1, with multiple isoforms, regulates osteogenesis via the p38 MAPK pathway, influencing *RUNX2* expression. Its inhibition reduces ALP activity, mineralization, and osteogenic markers (BMP2, collagen type I, osteocalcin) [57]. DDR2, in contrast, regulates *RUNX2* and *Osterix*, promoting extracellular matrix synthesis. Additionally, DDR2 directly contributes to osteoblast proliferation, matrix interaction, and early differentiation. Studies on DDR2-deficient mice highlight its critical role in skeletal growth and bone formation [58]. Despite both being collagen activated, they differ in expression patterns and regulatory mechanisms [59]. Our results show that B-Ex significantly upregulated these receptors, particularly DDR1, during early osteogenic differentiation of PL-MSCs, underscoring its role in initiating osteogenic transcription.

FOSB, a key AP-1 transcription factor, regulates osteoblast differentiation by promoting osteogenic genes such as *COL1A1*, *ALP*, and *BGLAP.* FOSB is critical for bone formation and adaptation to mechanical loading because it is induced by mechanical stress and growth factors and includes BMP and Wnt signaling [60]. Its overexpression in transgenic mice induces osteoblastic bone tumors, highlighting its pivotal role in bone biology. We observed that B-Ex increased FOSB expression, but more strongly in the late phase, mirroring its pattern under optimal osteogenic conditions and reinforcing its role in bone mineralization.

FOSL2 (Fra-2) is another AP-1 family transcription factor with distinct functions in bone biology. In contrast to FOSB, FOSL2 plays a distinct role in later osteoblast differentiation and extracellular matrix production, particularly type I collagen synthesis and osteoblast–osteoclast communication. It has been implicated in bone mass maintenance, with FOSL2 deficiency leading to altered bone remodeling [61]. Higher B-Ex concentrations elevated *FOSL2* expression in both basal and suboptimal conditions during the late differentiation phase, aligning with previous findings. However, its suppression under optimal osteogenic conditions remains unclear, possibly serving to regulate matrix formation and osteoblast maturation once these processes are fully established.

The mechanisms by which Biodentine and other tricalcium silicate cements promote osteoblastic differentiation of MSCs remain under investigation. Current evidence suggests that Ca released from Biodentine plays a key role in cytotoxicity, proliferation, and osteoinductive potential in MSC cultures [5]. A relevant comparison can be drawn with chitosan–TiO_2_ nanotube scaffolds incorporating Ca^2^^+^ ions. Lim et al. [62] found that Ca^2^^+^ concentrations up to 4.5 mM had minimal impact on osteoblast proliferation, whereas 13.5 mM stimulated growth. However, levels above 40 mM were cytotoxic, consistent with findings by Lee et al. [63], who observed optimal MSC proliferation at 6 mM. Similarly, we found that treating PL-MSCs with 6.2 mM Ca (concentration in 30% B-Ex) enhanced proliferation and *RUNX2* expression. Notably, these effects occurred even without additional osteoinductive factors.

To assess whether Ca from B-Ex mediated these effects, we treated B-Ex with EGTA, a calcium chelator. While the EGTA in our experimental setup fully suppressed Ca-induced proliferation and *RUNX2* expression, it did not completely inhibit B-Ex-induced *RUNX2* expression. This suggests that other B-Ex components, such as Si ions, phosphates, Biodentine microparticles, or hydroxyapatite, may contribute. Both phosphates and hydroxyapatite activate osteoblastic genes via the Erk1/2 pathway, with phosphates interacting with extracellular phosphate co-transporters and hydroxyapatite additionally engaging FGFR [64]. Several calcium channels facilitate cellular Ca entry, but the calcium-sensing receptor (CaSR) appears to play the most crucial role [65]. Therefore, further studies on this topic present a significant challenge.

In conclusion, our results demonstrate that B-Ex, at non-cytotoxic concentrations, enhances the proliferation and osteoblastic differentiation of PL-MSCs. This effect is further amplified when cells are co-stimulated with a suboptimal osteogenic medium. Notably, the kinetics and levels of osteoblastogenesis-related gene expression in B-Ex-treated cultures differ from those observed in cultures stimulated with the complete osteogenic medium, which served as a positive control. Overall, transcriptional dynamics suggest a prolonged osteoblastic differentiation in response to B-Ex. However, the process does not appear to have compromised final mineralization, as indicated by Alizarin Red staining. Using EGTA, a calcium chelator, we confirmed that PL-MSC proliferation and osteogenic differentiation, at least as indicated by *RUNX2* expression, are predominantly driven by elevated Ca levels in B-Ex. A limitation of this study is that it does not fully elucidate the mechanisms by which Ca, and potentially other factors in the extract, contribute to osteogenic differentiation. Nevertheless, our findings provide a strong foundation for further research, including the design of a clinical trial, particularly given the potential of PL-MSCs as a model for studying post-surgical healing of PLs following apicoectomy and root-end filling.

## 4. Materials and Methods

### 4.1. Periapical Tissue

Human PLs (*n* = 4) were obtained from four patients undergoing apicoectomy or tooth extraction at the Clinic for Stomatology, Military Medical Academy (MMA), Belgrade, Serbia. The study was approved by the MMA Ethical Committee (No. 1/2019) and conducted in compliance with the ethical principles outlined in the Declaration of Helsinki. Written informed consent was obtained from all participants. Exclusion criteria encompassed patients with cancer, autoimmune disorders, or diabetes, as well as those undergoing immunosuppressive treatment or having taken antibiotics within the past two weeks. PLs were diagnosed based on standard clinical and radiographic criteria. No distinction was made regarding patients’ age, sex, tooth type, lesion size, or clinical presentation of PLs. Two patients presented with two PLs on separate teeth, while the other two had a single lesion on one tooth. After extraction, PLs were immediately placed in RPMI-1640 medium (Sigma-Aldrich, Munich, Germany) supplemented with antibiotics and antimycotics and swiftly transported to the laboratory.

### 4.2. Preparation of Conditioned Medium from Biodentine

Biodentine (Septodont, Saint-Maur-des-Fossés, France) was aseptically prepared according to the manufacturer’s instructions. The polymerized material was pressed into 24-well plates, forming circular samples with a diameter of 10 mm and a thickness of 2 mm. These samples were then left under sterile laminar airflow for one hour to dry and complete polymerization. To obtain the eluate, referred to as Biodentine extract (B-Ex), each Biodentine disc was placed in a plastic tube containing RPMI-1640 medium supplemented with antibiotics and antimycotics. The Biodentine mass-to-medium ratio was maintained at 0.2 g/mL (2.2 cm^2^/mL), following ISO 10993 standards (ISO 10993:12-2021 [66]). The conditioning process lasted for three days, during which microparticles remained suspended in the medium, and the pH increased to 11.5. Before supplementing B-Ex with 10% fetal calf serum (FCS), its pH was adjusted to 7.4 using HCl. The prepared B-Ex was subsequently used to treat PL-MSCs.

### 4.3. Establishment of PL-MSCs

Cells were isolated from PLs following apicoectomy or tooth extraction. Immediately after extraction, the PLs were transferred to bottles containing α-MEM culture medium (Sigma-Aldrich, Darmstadt, Germany) supplemented with a cocktail of antibiotics and antimycotics (Sigma-Aldrich, Darmstadt, Germany) and promptly transported to the laboratory. The PL tissue was finely minced with scissors and enzymatically digested using collagenase type II (5 µg/mL) and DNAse (40 IU/mL) in serum-free α-MEM for one hour at 37 °C in an incubator with 5% CO_2_. Both enzymes were sourced from Sigma-Aldrich, Darmstadt, Germany.

The digested tissue was filtered through a 30 µm nylon mesh, rinsed with α-MEM, and centrifuged at 1800 rpm for 10 min. The resulting cell suspension was seeded into 24-well culture plates (Sarstedt, Nümbrecht, Germany) at a density of 2000 cells per cm^2^ and cultured in a complete medium optimized for MSCs. This medium consisted of α-MEM, 10% fetal calf serum (FCS) (Thermo Fisher Scientific, Grand Island, NY, USA), 100 IU/mL penicillin, 50 μg/mL streptomycin, 2.5 μg/mL amphotericin B (Thermo Fisher Scientific, Dreieich, Germany), 1% sodium pyruvate, and 100 µM L-ascorbate-2-phosphate (Sigma-Aldrich, Darmstadt, Germany).

After three days, non-adherent cells were removed by washing with α-MEM, and the medium was replaced twice weekly. Once the cells reached confluency, they were detached using 0.02% trypsin/0.02% NaEDTA (Sigma-Aldrich, Darmstadt, Germany) in phosphate-buffered saline (PBS) (Sigma-Aldrich, Darmstadt, Germany). The harvested cells were reseeded into 6-well plates at a density of 5000 cells per cm^2^ in the complete α-MEM medium, excluding amphotericin B. These cells, designated as PL-MSCs, were used in experiments after the fourth passage.

The colony-forming-unit fibroblast (CFU-F) assay was performed as follows: PL-MSCs were plated at a low density (200 cells per well in a 6-well plate) in the complete MSC culture medium. The cells were cultured for 10 days at 37 °C with 5% CO_2_, with the medium replaced every 2–3 days. At the end of the culture period, the cells were washed with PBS, fixed with 4% paraformaldehyde for 10–15 min, stained with hematoxylin-eosin, and rinsed with water. Colonies containing ≥ 50 cells were counted under a microscope to assess PL-MSC proliferation and clonogenic potential.

### 4.4. Phenotypic Analysis of PL-MSCs

After four passages, PL-MSCs were washed with PBS supplemented with 2% FCS and 0.1% sodium azide and then incubated in PBS/FCS/Na-azide containing primary antibodies conjugated to fluorescein isothiocyanate (FITC) or phycoerythrin (PE). The antibodies were used at the manufacturer-recommended dilution and incubated for 45 min at 4 °C. The following monoclonal antibodies were obtained commercially: anti-CD146-FITC, anti-CD90-FITC, anti-CD105-FITC, anti-CD166-FITC, and anti-CD56-PE from Serotec, Oxford, UK; anti-CD73-PE from R&D Systems, Minneapolis, MN, USA; and anti-STRO-1-FITC from Millipore/Chemicon, Billerica, MA, USA. Anti-CD39-PE and anti-SSA4-FITC antibodies were from Thermo Fisher Scientific, Waltham, MA, USA. Additionally, anti-CD45-FITC, anti-CD34-PE, anti-CD14-FITC, anti-CD3-PE, and anti-CD19-FITC were purchased from BioLegend, San Diego, CA, USA.

After antibody incubation, the cells were washed with PBS/Na-azide and analyzed using a CellFlow CUBE6 flow cytometer (Sysmex Partec GmbH, Görlitz, Germany). Results are expressed as the percentage of positive cells and mean fluorescence intensity (MFI) based on the analysis of at least 5000 cells per sample. Data were processed using FCS Express 6 RUO Edition software (version 6.00.0053).

### 4.5. Differentiation of PL-MSCs

The differentiation of PL-MSCs into osteoblasts was induced by culturing the cells in the osteoblast differentiation medium for 21 days in 6-well plates. Initially, the cells were cultured for 2 days in basal αMEM medium until they reached approximately 70% confluence. The medium was then replaced with a commercial osteoblast differentiation medium (Lonza Group Ltd., Basel, Switzerland), and culture continued for the next 21 days. The medium was refreshed every 4 days. After the incubation period, the cultures were washed and stained with Alizarin Red (Sigma-Aldrich, Darmstadt, Germany) to identify mineralized nodules.

For adipogenic differentiation, PL-MSCs were cultured in the adipogenic medium (Lonza Group Ltd., Basel, Switzerland) following the same procedure as for osteogenic differentiation, except that the culture period lasted 14 days. Adipocytes were identified by staining with Oil Red O (Sigma-Aldrich, Darmstadt, Germany).

For chondrogenic differentiation, the cells were centrifuged in polypropylene tubes, and the supernatant was discarded. A commercial chondrogenic medium (Lonza Group Ltd., Basel, Switzerland) was added to the pellet. The cells were cultured for 4 weeks, with medium changes twice a week. After the differentiation period, the pellet was embedded in a freezing medium (Bio-Optica Milano S.p.A, Milano, Italy) and frozen in liquid nitrogen. For analysis under a light microscope, 7 μm thick samples were cut on a cryotome (Leica Cryocut 1850, Leica Biosystems, Deer Park, IL, USA) and stained with Alcian Blue solution (Sigma-Aldrich, Darmstadt, Germany), which stains glycosaminoglycans, a component of the extracellular matrix produced by chondroblasts.

### 4.6. Cytotoxicity Tests

Cytotoxicity was evaluated according to ISO 10993-5:2009 [28], the standard biocompatibility test for medical devices. The measured variables included cell viability and metabolic activity of PL-MSCs. Viability was assessed based on the percentage of necrotic cells stained with propidium iodide (PI) (Sigma-Aldrich, Darmstadt, Germany), while metabolic activity was measured using the MTT assay, which evaluates succinate dehydrogenase activity in viable cells. The intensity of the formazan product was proportional to cell viability.

PL-MSCs were cultured in 96-well plates (1–2 × 10^4^ cells/well) in complete α-MEM medium. The next day, the medium was replaced with either fresh medium (control) or medium supplemented with different concentrations of B-Ex. PL-MSC cultures were incubated for 24 or 72 h. After incubation, the plates were centrifuged, the medium was carefully aspirated, and 100 µL of MTT solution (100 µg/mL) was added to each well. The plates were incubated for 4 h at 37 °C, after which the formazan crystals were dissolved overnight using 0.1 N HCl/10% sodium dodecyl sulfate (SDS) (Sigma-Aldrich, Darmstadt, Germany). Optical density (OD) was measured at 570/650 nm in an ELISA reader (ThermoFisher Scientific, Waltham, MA, USA), and results were expressed as relative metabolic activity compared with control cultures (100%). Wells without cells served as blank controls.

A parallel set of cultures was used to assess cell viability, calculated based on the percentage of necrotic (PI^+^) cells. After incubation, PL-MSCs were detached using 0.25% trypsin/0.2 mM EDTA (both from Sigma-Aldrich, Darmstadt, Germany), transferred to flow cytometry tubes, washed with PBS, and stained with PI (1 µg/mL) for 10 min in the dark. The cells were analyzed using a CellFlow CUBE6 flow cytometer (Sysmex Partec GmbH, Görlitz, Germany). Results were expressed as the percentage of PI^+^ cells, with viability calculated as follows: Viability (%) = 100% − % PI^+^ cells.

### 4.7. Modulatory Effect of Biodentine Extract on Osteoblast Differentiation of PL-MSCs

The effect of non-cytotoxic concentrations of B-Ex (30% and 10%) on the osteoblast differentiation of PL-MSCs was assessed in cultures maintained in basal medium alone, complete osteogenic medium, and basal medium supplemented with 30% osteogenic medium, referred to as suboptimal osteogenic medium. Osteoblast differentiation was evaluated in two ways: Gene expression analysis of osteoblastogenesis-related genes was performed on days 7 and 16. The mineralization index was assessed using Alizarin Red staining after 21 days of cultivation. Cultures were photographed using an Olympus inverted microscope equipped with a digital camera.

Mineralization scoring was performed semiquantitatively at 10× magnification based on the following criteria: Index 0: No visible positivity; Index 1: Mild staining of individual cells and the presence of 1–2 small mineralized nuclei stained pink-red in 1 of 10 analyzed fields; Index 2: Mild staining of individual cells and the presence of up to 5 small to medium-sized mineralized nuclei stained pink-red in 2 of 10 analyzed fields; Index 3: Presence of up to 10 mineralized nuclei of different sizes, stained pink-red, in 5 of 10 analyzed fields; Index 4: Presence of mineralized nuclei of all sizes, some fused, stained pink-red in all 10 analyzed fields.

### 4.8. Gene Expression Determination by Real-Time PCR

To determine the expression of genes involved in osteoblast differentiation of MSC, the Real-Time PCR method was used. The essential roles of 20 genes involved in osteoblast differentiation and bone metabolism are presented in Appendix A. Gene expression was determined in a set of parallel PL-MSC cultures as described above. Cells were collected after 7 days (early phase of osteoblastogenesis) and 16 days (late phase of osteoblastogenesis). Total RNA was isolated from the cell pellets, which were collected by centrifuging the PL-MSCs, using the RNeasy Mini Kit (Qiagen, Hilden, Germany). Complementary DNA (cDNA) was synthesized using a commercial reverse transcription kit following the manufacturer’s instructions (High-Capacity cDNA Reverse Transcription Kit; ThermoFisher Scientific, Waltham, MA, USA). qPCR of cDNA was performed using SYBR™ Green PCR Master Mix (2×, Thermo Fisher Scientific) with primer pairs specific for each of 20 genes of interest, synthesized by Microsynth, Balagach, Switzerland, and listed in Table 3. The entire procedure was carried out on the 7500 real-time PCR system (Applied Biosystems, Waltham, MA, USA). The qPCR reaction started with an initial denaturation step for 10 min at 95 °C, followed by a cycle consisting of a DNA strand denaturation step for 15 s at 95 °C, and a primer binding and strand elongation step for 1 min at 60 °C. The cycle was repeated 40 times.

The gene expression level of the gene of interest was standardized against the expression of the control gene, Glyceraldehyde-3-Phosphate Dehydrogenase (*GAPDH*), detected in the same sample and expressed as 2^−ΔCt^, where ΔCt is the difference between the Ct values of the gene of interest and β-actin. The relative expression of the gene of interest was later normalized as the relative increase or decrease (fold change) compared with the gene expression in the corresponding control (set as index 1). Each sample was analyzed in duplicate. The total number of analyses was four (two samples of two cell lines).

### 4.9. Proliferation Tests

The proliferation of PL-MSCs under the influence of different concentrations of B-Ex was primarily assessed using the MTT assay, as described in the cytotoxicity section. An additional test was performed based on the incorporation of [^3^H]-thymidine. In this assay, PL-MSCs were cultured in 96-well polystyrene plates in triplicate in the presence of different dilutions of B-Ex for 3 days. Proliferation was evaluated based on either the metabolic activity of the cultures (via MTT) or the amount of incorporated [^3^H]-thymidine (Amersham, Buks, UK; 1 µCi/mL of culture) compared with the control (cell proliferation in the control medium). Radioactive thymidine was added to the cultures during the final eight hours of incubation, and its incorporation was measured using a standard procedure with a scintillation β-counter (RAGBETA, Turku, Finland). The results were expressed as counts per minute (cpm).

### 4.10. The Role of Ca in the Proliferation and Osteoblastogenic Activity of Biodentine Extract

To investigate the role of Ca in PL-MSC proliferation and osteoblastic differentiation induced by B-Ex, the Ca-chelating agent EGTA was used. PL-MSCs cultured in the complete basal medium were treated with either 30% B-Ex alone or 30% B-Ex supplemented with 2 mM EGTA (Sigma-Aldrich, Darmstadt, Germany) for 24 h. After incubation, both the extract and the extract/EGTA were carefully washed away, and the cells were further cultured for an additional 48 h. Cell proliferation and *RUNX2* expression were then assessed. A parallel set of cultures was treated with 6.2 mM Ca (corresponding to the concentration determined in 30% B-Ex), with and without EGTA, as a specific control.

### 4.11. Statistics

The normality of the data was tested using the Shapiro–Wilk test. To assess differences between the parameters of control and experimental cultures, one- or two-way ANOVA with Tukey multiple comparison was used. Differences in mRNA expression between groups were analyzed using a ratio-paired *t*-test or Wilcoxon test. Values at *p* < 0.05 or less were considered to be statistically significant. The statistical analysis and graphs were performed in GraphPad Prism version 8.0.0 (GraphPad Software, San Diego, CA, USA).

## Figures and Tables

**Figure 1 ijms-26-04220-f001:**
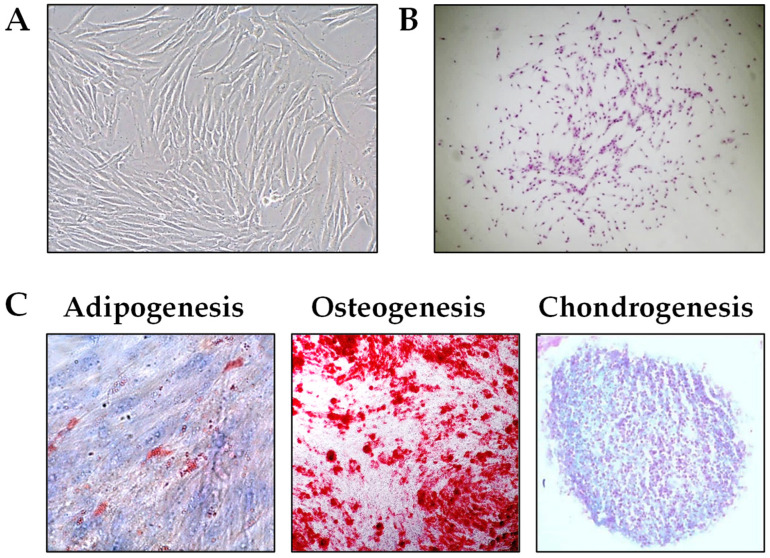
Fibroblastoid morphology (**A**), clonogenicity (**B**), and differentiation potential of PL-MSCs into adipocytes, osteoblasts, and chondroblasts (**C**). Magnifications: (**A**) ×200; (**B**) ×100; (**C**) ×400 (Adipogenesis); ×200 (Osteogenesis); ×100 (Chondrogenesis).

**Figure 2 ijms-26-04220-f002:**
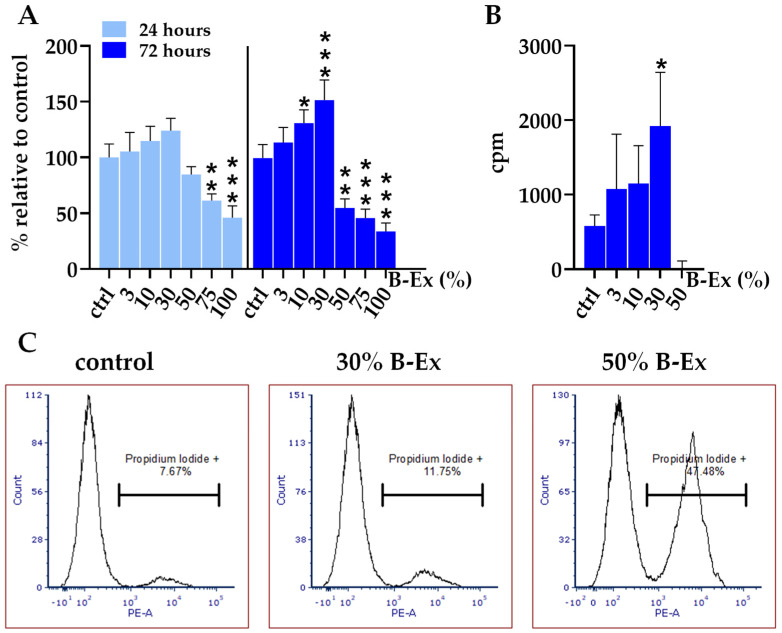
The effect of B-Ex on the metabolic activity (**A**), proliferation (**B**), and necrosis (**C**) of PL-MSCs. PL-MSCs were treated with different concentrations of B-Ex, as described in Section 4. Metabolic activity was assessed using the MTT assay after 24 and 72 h, while cell proliferation was evaluated by [^3^H]-thymidine incorporation after 72 h. Cell necrosis was determined by propidium iodide (PI) staining after 72 h. Data are presented as the mean ± SD of triplicate cultures ((**A**) and (**B**)) or as representative histograms using control, non-cytotoxic concentration of B-Ex (30%), and cytotoxic concentration (50% B-Ex) (**C**). * *p* < 0.05; ** *p* < 0.01; *** *p* < 0.001 compared with the corresponding controls (ctrl).

**Figure 3 ijms-26-04220-f003:**
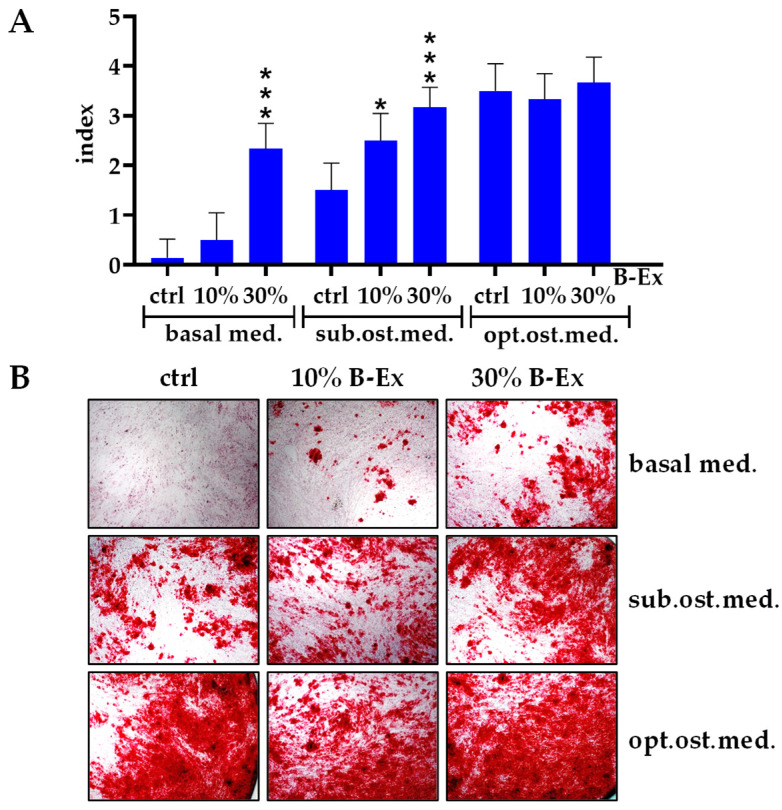
The effect of B-Ex on osteogenic differentiation of PL-MSCs as analyzed by Alizarin Red staining. PL-MSCs were cultivated in basal, suboptimal osteogenic, or optimal osteogenic medium with 10% or 30% B-Ex or without B-Ex (control) as described in Section 4. Results are presented as mineralization indexes (**A**) or representative images (**B**) Magnifications ×200. * *p* < 0.05; *** *p* < 0.001 compared with the corresponding controls (ctrl).

**Figure 4 ijms-26-04220-f004:**
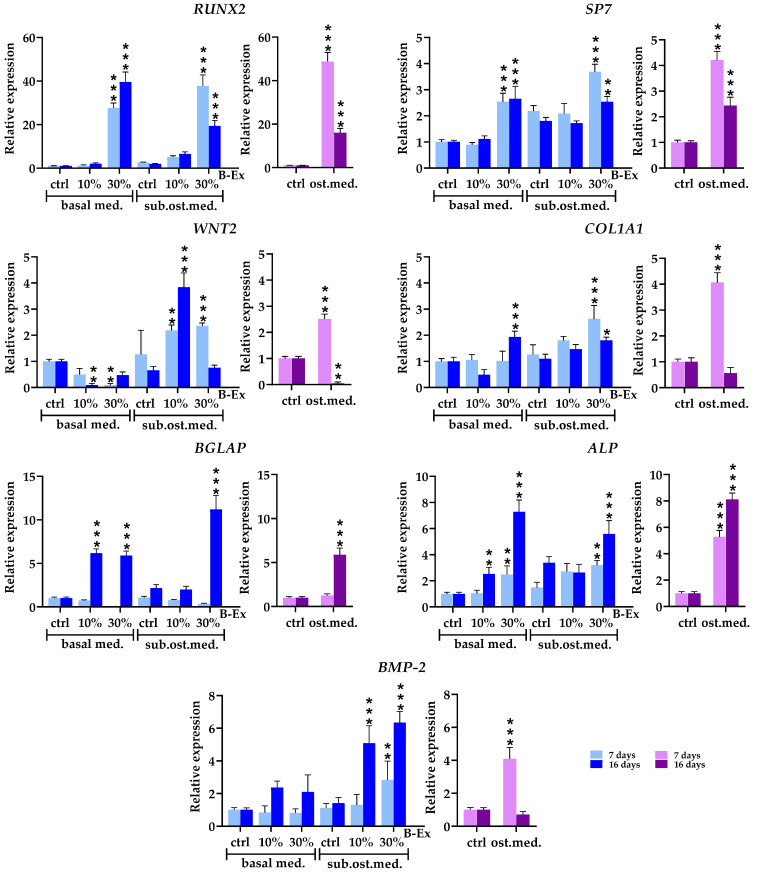
The effect of B-Ex on the expression of key osteoblastic genes in PL-MCSs. The methodology of the experiment is described in Section 4. Data are presented as mean ± SD (*n* = 4) of relative mRNA expression compared with control PL-MSCs used as index 1. * *p* < 0.05; ** *p* < 0.01; *** *p* < 0.001 compared with the corresponding controls (ctrl).

**Figure 5 ijms-26-04220-f005:**
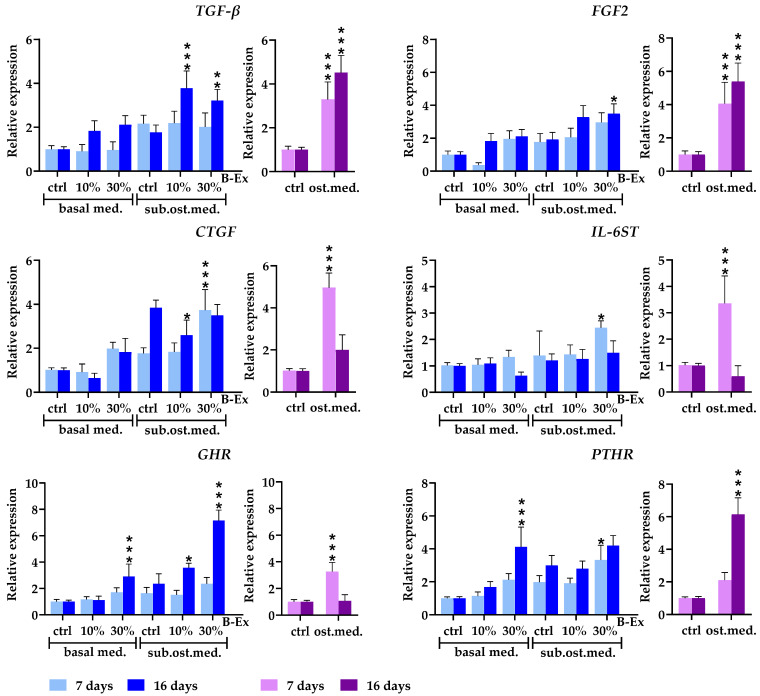
The effect of B-Ex on the expression of cytokine/cytokine receptor and hormone receptor genes in PL-MCSs. The methodology of the experiment is described in Section 4. Data are presented as mean ± SD (*n* = 4) of relative mRNA expression compared with control PL-MSCs, used as index 1. * *p* < 0.05; ** *p* < 0.01; *** *p* < 0.001 compared with the corresponding controls (ctrl).

**Figure 6 ijms-26-04220-f006:**
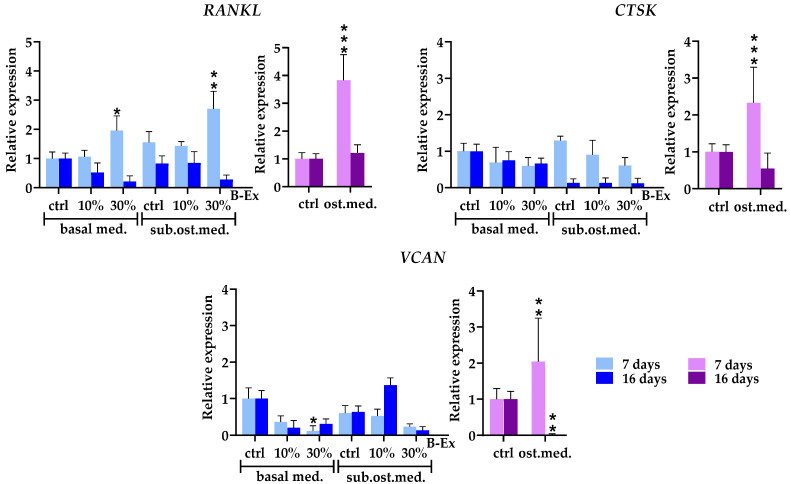
The effect of B-Ex on the expression of genes modeling the bone and extracellular matrix in PL-MCSs. The methodology of the experiment is described in Section 4. Data are presented as mean ± SD (*n* = 4) of relative mRNA expression compared with control PL-MSCs, used as index 1. * *p* < 0.05; ** *p* < 0.01; *** *p* < 0.001 compared with the corresponding controls (ctrl).

**Figure 7 ijms-26-04220-f007:**
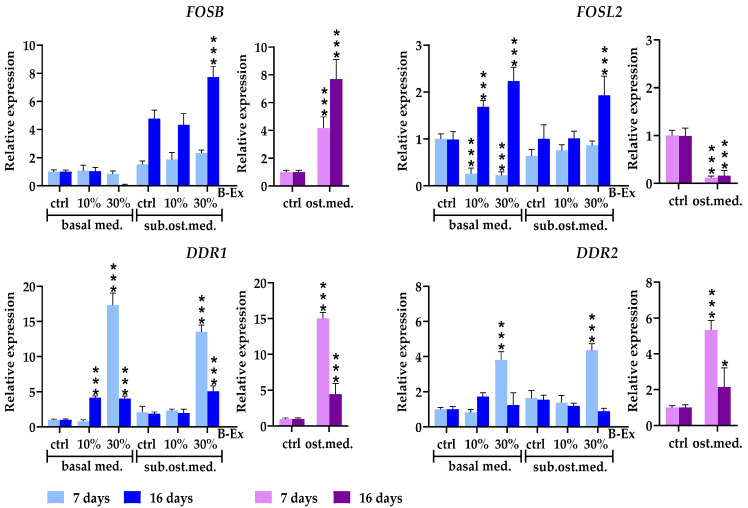
The effect of B-Ex on the expression of genes modulating osteoblast signaling in PL-MCSs. The methodology of the experiment is described in Section 4. Data are presented as mean ± SD (*n* = 4) of relative mRNA expression compared with control PL-MSCs, used as index 1. * *p* < 0.05; *** *p* < 0.001 compared with the corresponding controls (ctrl).

**Table 1 ijms-26-04220-t001:** Phenotypic analysis of PL-MSCs.

Markers	% Positive Cells
CD73	99.43 ± 0.34
CD90	99.52 ± 0.30
CD105	91.23 ± 3.24
CD166	99.02 ± 0.28
CD39	90.50 ± 3.64
CD146	28.18 ± 6.25
CD56	20.04 ± 4.88
SRTRO-1	8.06 ± 4.60
SSEA4	12.12 ± 5.56
CD45	1.12 ± 0.46
CD34	1.24 ± 0.37
CD14	1.50 ± 0.24
CD3	0.54 ± 0.14
CD19	0.48 ± 0.22

The phenotypic analysis of the PL-MSC cell lines after four passages in culture was performed as described in Section 4. The results represent the mean percentage of positive cells from three different PL-MSC lines ± SD.

**Table 2 ijms-26-04220-t002:** The effect of EGTA pretreatment on proliferation and *RUNX2* expression in PL-MSC cultures stimulated with Ca or B-Ex.

Culture Conditions	Proliferation (cpm)	RUNX2 Expression
Negative control	642 ± 88	1.00 ± 0.12
Ca	1864 ± 198 ***	16.34 ± 1.92 ***
Ca + EGTA	706 ± 102 ^●●^	0.92 ± 0.04 ^●●●^
30% B-Ex	2122 ± 170 ***	22.6 ± 3.90 *** #
30% B-Ex + EGTA	734 ± 102 ^●●●^	3.70 ± 0.86 *** ^●●●^

PL-MSCs were pretreated with CaCl_2_ (6.2 mM Ca) or 30% B-Ex, with or without 2 mM EGTA, for 24 h, followed by washing and further culture in the complete basal medium for an additional 48 h. Results are presented as mean ± SD (*n* = 3). *** *p* < 0.001 vs. the negative control. ^●●^
*p* < 0.01; ^●●●^
*p* < 0.001 vs. corresponding controls (without EGTA). # *p* < 0.05 vs. Ca-supplemented cultures.

**Table 3 ijms-26-04220-t003:** Sequences of the primer pairs used for the real-time PCR experiments.

Primers	Sequences (5′-3′)
*WNT2*	TTCCAGAGCTAACTCGTGCC ACTGGGCTTGAAGGGTGATG
*COL1A1*	TCGGAGGAGAGTCAGGAAGG AACAGAACAGTCTCTCCCGC
*SP7*	TTCTGCGGCAAGAGGTTCACTC GTGTTTGCTCAGGTGGTCGCTT
*BMP-2*	GGGGTGGGGGAAAGGTAATG TCGGGTTATCCAGGTTTTGCT
*RUNX2*	GCGGTGCAAACTTTCTCCAG TCACTGTGCTGAAGAGGCTG
*BGLAP*	GACTGTGACGAGTTGGCTGA CACATCCATAGGGCTGGGAG
*TGF-β1*	GGACACCAACTATTGCTTCAGCTCC AGGCTCCAAATGTAGGGGCAGGGCC
*FGF 2*	GGGTGCCAGATTAGCGGAC GTTCACGGATGGGTGTCTCC
*ALP*	GGGCATTGTGACTACCACTC AGTCAGGTTGTTCCGATTCA
*FOSB*	TCTGTCTTCGGTGGACTCCTTC GTTGCACAAGCCACTGGAGGTC
*FOSL2*	AAGAGGAGGAGAAGCGTCGCAT GCTCAGCAATCTCCTTCTGCAG
*DDR1*	GCGTCTGTCTGCGGGTAGAG ACGGCCTCAGATAAATACATTGTCT
*DDR2*	TGTTCCTGCTGCTGCCTATCTT AGGATAGCGGCATATAGCTGGAT
*CTGF*	GTTTGGCCCAGACCCAACT GGAACAGGCGCTCCACTCT
*VCAN*	TTGGACCTCAGGCGCTTTCTAC GGATGACCAATTACACTCAAATCAC
*RANKL*	GCCTTTCAAGGAGCTGTGCAAAA GAGCAAAAGGCTGAGCTTCAAGC
*CTSK*	GAGGCTTCTCTTGGTGTCCATAC TTACTGCGGGAATGAGACAGGG
*IL-6ST*	CACCCTGTATCACAGACTGGCA TTCAGGGCTTCCTGGTCCATCA
*PTHR*	CATCTTTTGGTCCATCTGTCCATC CTGGAGACCCTCGAGACCACA
*GHR*	TGCTTTTTCTGGAAGTGAGGA GGTTCTTTGTACCATGATGAACCT
*GAPDH*	GTCTCCTCTGACTTCAACAGCG ACCACCCTGTTGCTGTAGCCAA

## Data Availability

All data are included in this article.

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
