# Peer review of "Biodentine Stimulates Calcium-Dependent Osteogenic Differentiation of Mesenchymal Stromal Cells from Periapical Lesions"

_ijms, 2025, doi:10.3390/ijms26094220_

Round 1
Reviewer 1 Report
Comments and Suggestions for Authors
In the manuscript “Biodentine stimulates calcium-dependent osteogenic differentiation of mesenchymal stromal cells from periapical lesions,” the authors present an interesting and novel study that demonstrates the influence of biodentine composition on the expression of various osteoblastogenesis-related genes and the effect of the presence of the ion calcium in stimulating proliferation and the phenotype lineage of osteoblasts by analysis the gene expression.
The paper can be accepted for publication after a minor revision, considering the following comments:
- The authors must specify that “Biodentine” is the registered trademark of the tricalcium silicate cement evaluated in this study.
- The authors must specify why they chose only one cell line for all assays or explain the basis for selecting only one cell line, rather than all four cell lines mentioned in section 2.1, lines 82-91.
- In Lines 128-132 of section 2.2, the figure references are confusing because the propidium iodide (PI) staining results correspond to Fig. 2C and not to Fig. 2B.
- The explanation of Fig. 2C for 50% B-Ex was not included.
- In Section 2.3, it is suggested to indicate which information corresponds to Fig. 3A and Fig. 3B.
- The authors could include a section on the disadvantages of Biodentine, which must be discussed.
- In Section 4.2, why was Biodentine without extraction not considered in the study?
Author Response
Comments and Suggestions for Authors
In the manuscript “Biodentine stimulates calcium-dependent osteogenic differentiation of mesenchymal stromal cells from periapical lesions,” the authors present an interesting and novel study that demonstrates the influence of biodentine composition on the expression of various osteoblastogenesis-related genes and the effect of the presence of the ion calcium in stimulating proliferation and the phenotype lineage of osteoblasts by analysis the gene expression.
The paper can be accepted for publication after a minor revision, considering the following comments:
- The authors must specify that “Biodentine” is the registered trademark of the tricalcium silicate cement evaluated in this study.
Reply: I added this in the first sentence of Introduction
- The authors must specify why they chose only one cell line for all assays or explain the basis for selecting only one cell line, rather than all four cell lines mentioned in section 2.1, lines 82-91.
Reply: Based on our initial experiments one cell line was sufficient to screen cytotoxicity to determine non-cytotoxic concentrations of substances or extracts, because the responses of different lines are very uniform. Typically, three lines are required to assess marker expression as phenotypic variations serve as a sensitive indicator of interline variability. We chose to use two lines with very similar phenotypes to study gene expression in duplicates. Since the results were highly consistent, further testing was unnecessary. The role of Ca was examined in a single line with a sufficient number of replicates, as the effect was compelling, making additional testing on multiple lines unnecessary.
- In Lines 128-132 of section 2.2, the figure references are confusing because the propidium iodide (PI) staining results correspond to Fig. 2C and not to Fig. 2B.
Reply: Our results are presented correctly. As indicated, Figure 2B refers to cell proliferation, determined by ³H-thymidine incorporation and expressed in cpm. Figure 2C refers to necrosis, as determined by PI staining.
- The explanation of Fig. 2C for 50% B-Ex was not included.
Reply: We believe the presentation is clear; however, we have added an additional explanation in the Figure 2 legend (marked in red).
- In Section 2.3, it is suggested to indicate which information corresponds to Fig. 3A and Fig. 3B.
Reply: This is added now (marked in red)
- The authors could include a section on the disadvantages of Biodentine, which must be discussed.
Reply: We added a sentence in the Introduction as follows: Although larger clinical studies are required to evaluate the effectiveness of Biodentine in root-end filling and potential disadvantages, such as difficulty in initial handling [14], the experience of the apical surgeon is of crucial importance (lines, 77-80).
- In Section 4.2, why was Biodentine without extraction not considered in the study?
Reply: The reason for using the extract is sufficiently elaborated in the Introduction (lines, 89-93) and Discussion (lines, 312-316), based on the fact that after root-end filling, both microparticles and soluble products are released from Biodentine, and in this context, they may modulate periapical regenerative tissue. The cultivation of PL-MSCs directly on Biodentine discs was not the aim of our study due to higher cytotoxicity (our unpublished results) and because this model is not relevant for studying full osteogenic differentiation (Alizarin Red staining)
Reviewer 2 Report
Comments and Suggestions for Authors
Dear Authors,
After carefully reviewing the manuscript titled : “Biodentine Stimulates Calcium-Dependent Osteogenic Differentiation of Mesenchymal Stromal Cells From Periapical Lesions“ I can conclude that topic is interesting for readers, however I have some questions/suggestions to clarify.
The abstract is well-structured, but it could be more concise, especially part regarding methods.
The introduction provides a solid background on Biodentine but could better highlight the facts why this study´s focus on PL-MSCs is novel, and why existing studies on Biodentine are insufficient.
In materials and methods I suggest to specify if any microscopic or spectrophotometric analysis was conducted to confirm the composition of the extract before testing. The study evaluates 20 genes, but only some results are discussed in depth.
Consider adding a brief table summarizing key gene functions to improve readability.
In results section, figure 2A is informative but I suggest to indicate sample size (n) in figure legends to specify whether normality tests were conducted before applying statistical tests.
Discussion needs more comparison with previous studies. You should be more focused on how PL-MSC behavior differs from periodontal ligament stem cells (PDLSCs) or other MSCs used in prior studies. Explain how prolonged osteogenesis could impact bone healing post-apicoectomy. Is delayed differentiation beneficial or potentially problematic?
The conclusion summarizes findings well but does not propose future research directions. Add a sentence suggesting follow-up studies, such as in animal models or clinical trials, to validate Biodentine’s osteogenic effects in vivo.
Author Response
Comments and Suggestions for Authors
Dear Authors,
After carefully reviewing the manuscript titled : “Biodentine Stimulates Calcium-Dependent Osteogenic Differentiation of Mesenchymal Stromal Cells From Periapical Lesions“ I can conclude that topic is interesting for readers, however I have some questions/suggestions to clarify.
The abstract is well-structured, but it could be more concise, especially part regarding methods.
Reply: This is now modified ( part of the methodology is omitted but the dominant aspect of results is outlined..marked in red in the abstract)
The introduction provides a solid background on Biodentine but could better highlight the facts why this study´s focus on PL-MSCs is novel, and why existing studies on Biodentine are insufficient.
Reply: the existing studies on Biodentine in other models have already been discussed. The novelty is clarified additionally in the Introduction (We believe this unique study model is suitable for drawing valid conclusions, as leachable components and microparticles of Biodentine during root-end filling could reach periapical tissues and affect MSC differentiation. Therefore, this in vitro model best mimics the process of osteogenic differentiation in vivo during periapical healing.) and Discussion (This original in vitro model mimics the impact of Biodentine’s released compounds on the osteoinductive potential of this population of inflammatory MSCs [13-15], which come into close contact in vivo with Biodentine and its components).
In materials and methods I suggest to specify if any microscopic or spectrophotometric analysis was conducted to confirm the composition of the extract before testing.
Reply: SEM, EDX, FTIR, and XRD characterization, as well as elemental ICP analysis of the extract, are currently in progress, and this is mentioned in the Discussion (lines, 338-340). For this study, the most relevant result is related to the Ca concentrations, and this has already been mentioned in the text."
The study evaluates 20 genes, but only some results are discussed in depth.
Consider adding a brief table summarizing key gene functions to improve readability.
Reply: We have extended the discussion on these genes while trying to keep the text as concise as possible (marked in red). The suggested table is presented as Supplementary Table 1.
In results section, figure 2A is informative but I suggest to indicate sample size (n) in figure legends to specify whether normality tests were conducted before applying statistical tests.
Reply: There is a clear indication that the samples are triplicates. 'Triplicate' means three. Yes, normality is always checked, and this is added in the subsection Statistics.
Discussion needs more comparison with previous studies. You should be more focused on how PL-MSC behavior differs from periodontal ligament stem cells (PDLSCs) or other MSCs used in prior studies.
Reply: As suggested we added detailed phenotypic comparisons with PDLSCs including 4 new references (lines, 326-328 ).
Explain how prolonged osteogenesis could impact bone healing post-apicoectomy. Is delayed differentiation beneficial or potentially problematic?
Reply: Prolonged osteogenesis did not impact significantly osteogenesis as judged by Alizarin Red staining. This is now added in the Discussion ( lines, 551-553)
The conclusion summarizes findings well but does not propose future research directions. Add a sentence suggesting follow-up studies, such as in animal models or clinical trials, to validate Biodentine’s osteogenic effects in vivo.
Reply: Added (line 558)
Round 2
Reviewer 2 Report
Comments and Suggestions for Authors
All requested changes were made and all questions were answered.